# Quantification of Structural Integrity and Stability Using Nanograms of Protein by Flow-Induced Dispersion Analysis

**DOI:** 10.3390/molecules27082506

**Published:** 2022-04-13

**Authors:** Morten E. Pedersen, Jesper Østergaard, Henrik Jensen

**Affiliations:** 1Fida Biosystems ApS, Generatorvej 6, 2860 Soeborg, Denmark; morten@fidabio.com; 2Department of Pharmacy, University of Copenhagen, Universitetsparken 2, 2100 Copenhagen, Denmark; jesper.ostergaard@sund.ku.dk

**Keywords:** FIDA, protein stability, protein size, hydrodynamic radius, Taylor dispersion analysis (TDA), protein folding, automation

## Abstract

In the development of therapeutic proteins, analytical assessment of structural stability and integrity constitutes an important activity, as protein stability and integrity influence drug efficacy, and ultimately patient safety. Existing analytical methodologies solely rely on relative changes in optical properties such as fluorescence or scattering upon thermal or chemical perturbation. Here, we present an absolute analytical method for assessing protein stability, structure, and unfolding utilizing Taylor dispersion analysis (TDA) and LED-UV fluorescence detection. The developed TDA method measures the change in size (hydrodynamic radius) and intrinsic fluorescence of a protein during in-line denaturation with guanidinium hydrochloride (GuHCl). The conformational stability of the therapeutic antibody adalimumab and human serum albumin were characterized as a function of pH. The simple workflow and low sample consumption (40 ng protein per data point) of the methodology make it ideal for assessing protein characteristics related to stability in early drug development or when having a scarce amount of sample available.

## 1. Introduction

The stability of therapeutic proteins is essential to efficacy and patient safety, and is thus an important product-quality attribute from early drug development to release testing of the final drug product [1,2,3]. Protein stability can be assessed via forced denaturation experiments involving thermal ramps or chemical denaturant titration series [4,5,6]. These assays primarily report a denaturation midpoint value expressed as the temperature (*T*_m_) or concentration (*C*_m_), leading to equal fractions of native and denatured protein [6]. This readout can be utilized for straightforward ranking of constructs as well as formulation condition screenings, including ionic strength, pH, and excipients. The analytical methodologies in this segment are typically confined to measure the relative change in fluorescence or heat during the forced denaturation process [4,7]. Thus, data on structural properties such as hydrodynamic radius must be obtained via orthogonal techniques e.g., dynamic light scattering (DLS) and small-angle x-ray scattering [5,8]. 

Recently, we presented a multi-tiered approach for assessing protein stability, structure, and function utilizing flow-induced dispersion analysis (FIDA) [9,10,11,12] and Taylor dispersion analysis (TDA) [13,14,15] with intrinsic fluorescence detection [16]. The primary advantages associated with the FIDA and TDA approaches are the provision of an additional level of information in terms of protein structure, in the form of hydrodynamic radius, as well as intrinsic fluorescence in a single measurement [16]. Furthermore, FIDA and TDA are performed in solution and are nondestructive towards the sample [16]. 

The aim of this work is to introduce an optimized TDA protocol using considerably less protein (40 ng of protein per measurement), as well as a simpler workflow as compared to the initial work [16]. The new approach is based on measuring the change in apparent hydrodynamic radii and intrinsic fluorescence of a protein as it undergoes mixing, and thus chemical denaturation, with guanidinium chloride (GuHCl) inside the flow channel. Preparation of tedious and manual dilution series are thus completely omitted. The method was developed while studying the stability towards unfolding of adalimumab and human serum albumin (HSA) at different pH values. Finally, we derived fitting models for quantitative determination of the protein-stability parameters *C*_m_ and Δ*G*° (Gibbs free energy of unfolding). 

## 2. Results and Discussion

### 2.1. Assay Development of Capillary Mixing Mode for Protein-Stability Assessment

The rapid and straightforward TDA method for assessing protein structure and conformational stability is schematically shown in Figure 1. Under the conditions of the TDA experiment, radial diffusion is dominating over diffusion in the length direction of the capillary. Therefore, fast-diffusing species will exhibit a narrow dispersion, whereas slow-diffusing species will lead to a broad dispersion. In detail, 40 nL of protein sample (corresponding to 40 ng protein per data point) is mixed with the denaturant guanidinium chloride (GuHCl) inside the capillary, and the denaturation process therefore takes place during the experimental time course, which is usually ~2–4 min. The mixing process is generated by the laminar flow and sample dispersion into the denaturant solution. Detection of the protein is performed by UV-LED fluorescence, which measures intrinsic fluorescence from the tryptophan and tyrosine residues in the proteins with nM-µM sensitivity. Therefore, the protein sample is measured without any chemical modification or introduction of tags (e.g., His, GFP) as well as under in-solution conditions. In addition to providing the taylorgram from which hydrodynamic radius can be extracted, the intrinsic fluorescence detection reveals structural features related to changes in fluorescence intensity from the taylorgram peak area. Representative taylorgrams for the unfolding of adalimumab at pH 10 are shown in Appendix A.

The apparent hydrodynamic radius (*R*_h_) and intrinsic fluorescence of the protein sample was measured throughout a titration series with increasing GuHCl concentrations (0–6 M). This formed the basis for the generation of a denaturation curve that serves as a measure of protein stability as well as determination of the denaturation midpoint (*C*_m_) representing the GuHCl concentration where half of the protein is unfolded. An evaluation of protein stability under different conditions such as pH, ionic strength, and addition of excipients can be conducted. In this work, we probe the stability of the therapeutic protein adalimumab as well as human serum albumin (HSA) as a function of pH.

### 2.2. Unfolding of HSA and Adalimumab Using Capillary Mixing Mode

The apparent hydrodynamic radii and intrinsic fluorescence intensities of HSA and adalimumab determined using TDA as a function of GuHCl concentration (0–6 M) at pH 4, 7, and 10 are shown in Figure 2. HSA exhibited an increase in hydrodynamic radius (*R*_h_) from ~3.9 to 6.0 nm for pH 7 and pH 10. In contrast, at pH 4 HSA showed a higher *R*_h_ of 4.4–4.8 nm at low GuHCL concentrations. This might be due to the presence of HSA dimers, which has been reported for serum albumin (BSA) at acidic pH [17]. A fluorescence signal was not detected for HSA in neat buffer (0 M GuHCl) at pH 4, most likely due to a higher degree of HSA protonation (pI = 4.7 [18]) and thus extensive surface adsorption of HSA to the inner surface of the capillary at this pH. A decrease in intrinsic fluorescence of HSA upon increasing GuHCl concentrations was observed (Figure 2). The increase in *R*_h_ and decrease in intrinsic fluorescence revealed HSA unfolding. An increase in the *R*_h_ of HSA upon denaturation is expected since the native and compact structure is unfolded, which also leads to a decrease in intrinsic fluorescence caused by solvent exposure of the tryptophan and tyrosine residues [16]. Figure 2 shows that the stability of HSA was similar at pH 7 and 10, whereas at pH 4, HSA was more stable towards GuHCl denaturation as seen by a right-shifted unfolding curve. The changes observed with respect to apparent *R*_h_ and intrinsic fluorescence upon HSA unfolding were in line with previous TDA experiments utilizing urea and a pre-incubation TDA setup [16]. This provided evidence to the suitability of the present capillary mixing approach. In comparison to the established pre-incubation methodology [16], the capillary mixing mode is advantageous in terms of simpler workflow and less protein per measurement (40 ng). 

The TDA-based capillary mixing method was applied for probing unfolding and stability of the therapeutic antibody adalimumab. Similar to HSA, the signal was not detected for adalimumab (pI = 8.5 [19]) in neat buffer (0 M GuHCl) at pH 4 and 7 due to surface adsorption, but in the presence of ≥0.5 M GuHCl, a signal due to the intrinsic fluorescence of adalimumab was detected. Adalimumab exhibited an increase in *R*_h_ from ~5.2 to 7.9 nm, which was accompanied with an increase in intrinsic fluorescence (Figure 2), which was opposite to the intensity changes observed during HSA unfolding. This trend has been described previously in the literature for antibodies upon unfolding [20], and can be ascribed to fluorescence quenching among tryptophan and tyrosine residues in proximity within the folded state. Adalimumab was least stable towards GuHCl-induced unfolding at pH 4, while at pH 7 and 10 adalimumab showed similar stability. The results of the adalimumab denaturation experiments were comparable with a TDA study showing an increase in adalimumab *R*_h_ (4.9–6.9 nm) upon thermal stress [21]. 

Each data point, in these sets of experiments, consumed only 40 ng of protein. In practice, 20 µg of protein (i.e., 20 µL of a 1 mg/mL protein solution) was applied per pH value since the capillary has to be physically submersed into the sample vial insert during the injection procedure. However, the remaining sample can be recovered from the vial after analysis. Furthermore, lowering the protein concentration to match the UV-LED detector limits will also considerably reduce protein consumption.

The developed method is applicable to proteins with tryptophan or tyrosine residues when employing UV-LED fluorescence detection. A requirement for the capillary mixing mode applied here is fast denaturation kinetics relative to the time required to mobilize the sample from the inlet end of the capillary to the detection window (the order of minutes). The apparent *R*_h_ of HSA at maximum denaturant concentration obtained with capillary mixing was similar to the *R*_h_ determined using pre-incubation conditions (see Appendix A). Krishnakumar and Panda reported that the unfolding kinetics of HSA at pH 7 is described by an exponential decay slowly approaching the equilibrium condition (minutes–hours) when utilizing intrinsic tryptophan fluorescence measurements and GuHCl as denaturant [22]. In that light, the developed TDA approach may report intermediate protein stability, since the experimental timeframe was a few minutes. The influence on HSA unfolding as a function of reaction time inside the capillary, as well as the applied mixing principle, was therefore studied (see Appendix A). These results showed small differences between pre-incubated samples and samples mixed in the capillary, indicating that unfolding takes place on a longer timescale than the standard mixing time in the capillary. Unfolding kinetics may be probed by changing the mobilization pressure.

At high GuHCl concentrations, several taylorgrams exhibited a local fluorescence dip or raise in the peak center, which was ascribed to buffer mismatch caused by the capillary mixing mode. Mitigation of this phenomenon by fitting the taylorgrams to a double Gaussian function is described in the Appendix A. 

Collectively, the results obtained for HSA and adalimumab unfolding in buffer solution as a function of GuHCl concentration and pH show that HSA is more prone to GuHCl-induced unfolding than adalimumab. Thus, changes in *R*_h_ and intrinsic fluorescence occurred at different GuHCl concentrations and pH. Interestingly, adalimumab was least stable at pH 4, whereas HSA exhibited its highest stability at this point, most likely due to dimerization at acidic pH [17]. Differences were also apparent for changes in intrinsic fluorescence upon unfolding between the two proteins, since it decreased for HSA and increased for adalimumab. Both observations were in line with previous studies in the literature [16,20,23], and therefore ascribed to differences in higher-order structure and the number of tryptophan and tyrosine residues.

### 2.3. Quantitative Comparison of Stability towards Unfolding

The denaturation curves were fitted to Equation (1) for calculating both the denaturation midpoint (*C*_m_) and standard free-energy change (Δ*G*° (H_2_O)) of the proteins (adalimumab and HSA) at pH 4, 7, and 10 (Table 1). The unfolding model (Equation (1)) utilizes the hydrodynamic radius (*R*_h_) as a function of denaturant concentration, assuming a reversible two-state model comprising folded and unfolded protein. The unfolding model is derived in the Appendix A. 

The *C*_m_-value represents the GuHCl concentration with equal fractions of folded and unfolded protein, and thus facilitates direct comparison of protein stability where a high *C*_m_-value corresponds to high stability towards unfolding. Table 1 demonstrates that the antibody adalimumab was generally more stable than HSA, except at pH 4 where the adalimumab was more prone to unfolding. The thermodynamic parameter, standard free-energy change (Δ*G*° (H_2_O)), describes the conformational stability of a protein in absence of denaturant at a given condition e.g., temperature, pH, and ionic strength [6]. Conditions leading to higher conformational stability yield higher Δ*G*°. Furthermore, the determined Δ*G*° can be utilized for revealing fractions of denatured and aggregated protein [24]. The thermodynamic parameters for HSA at pH 7.0 (Table 1) were similar to a study conducted by Farruggia et al. reporting Δ*G*° (H_2_O) of 3.4 ± 0.3 kcal/mol and *C*_m_ of 2.7 ± 0.1 M for HSA denaturation by GuHCl (pH 7.4, 20 °C) utilizing spectrofluorimetry [25]. Discrepancy in the determined Δ*G*° (H_2_O)’s may be ascribed to differences in measuring principles and experimental conditions such as pH, temperature, mixing time, and ionic strength. The influence on the ionic strength from using GuHCl as denaturant can be handled by applying alternative chemical denaturants such as urea or tetramethylurea.

## 3. Methods

### 3.1. Materials and Chemicals

#### Raw Materials

Adalimumab (Amgevita, Amgen Europe B.V., Breda, The Netherlands) was obtained from Nomeco (Copenhagen, Denmark). Human serum albumin (HSA), guanidine hydrochloride, L-histidine monohydrochloride monohydrate, sodium succinate dibasic, hydrochloric acid, and sodium hydroxide were obtained from Sigma-Aldrich (St. Louis, MI, USA). N-2-Hydroxyethylpiperazine-N′-2-ethanesulfonic acid (HEPES) was obtained from VWR Chemicals BDH (Leuven, Belgium). Ultrapure water (18.2 MΩ-cm at 25 °C) was obtained from a Direct-Q 3 (type 1) water-purification system (Merck KGaA, Darmstadt, Germany).

The assay buffer was a so-called broad-range buffer prepared with purified water, 20 mM HEPES, 20 mM histidine-HCl, and 20 mM sodium succinate following pH adjustment to pH 4.0, 7.0, or 10.0 with NaOH or HCl. The prepared buffer solutions were filtered through a 0.22 μm Q-Max filter (Frisenette, Knebel, Denmark).

### 3.2. Equipment

Experiments were performed on a Fida 1 instrument with a UV-LED fluorescence detector (Ex 275 nm/Em > 300 nm) from Fida Biosystems ApS (Copenhagen, Denmark). High-sensitivity coated capillaries with inner diameter 75 µm, outer diameter 375 µm, total length 100 cm, and length to detection window 84 cm, from Fida Biosystems were used for all experiments. 

### 3.3. Methods

#### 3.3.1. Procedure

Each measurement consumed 40 nL of sample (protein solution) with an analysis time of 6 min using the following stepwise procedure: Step 1, flush with assay buffer at 3500 mbar for 60 s; Step 2, flush with denaturant solution (0–6 M GuHCl) at 3500 mbar for 30 s; Step 3, injection of protein sample at 50 mbar for 10 s; Step 4, mobilize and measure with denaturant solution (0–6 M GuHCl) at 400 mbar for 280 s. The taylorgram was obtained in Step 4.

#### 3.3.2. Denaturation Curves

The hydrodynamic radii of 1 mg/mL adalimumab and 1 mg/mL HSA were measured separately in a titration series with denaturant (GuHCl, 0–6 M) at pH 4.0, 7.0, and 10.0, employing TDA with intrinsic fluorescence detection. For each experimental pH value, the proteins (adalimumab and HSA) were diluted from their respective stock solutions (adalimumab (50 mg/mL) and HSA (15 mg/mL)) with the corresponding assay buffer into a sample vial. Upon injection, the sample and denaturant were mixed inside the capillary during the mobilization period (~2–4 min) where the sample was moving towards the detector; this is the so-called capillary mixing mode. All samples were analyzed in triplicate.

#### 3.3.3. Unfolding Kinetics of HSA

The unfolding kinetics of 1 mg/mL HSA was studied at pH 7.0 by varying the mobilization pressure (i.e., procedural Step 4) between 50–800 mbar, which was equivalent to denaturation incubation times between 1–26 min inside the capillary (Appendix A). Furthermore, the previously reported premixing mode [16] was also applied in order to benchmark the incubation times against equilibrium conditions (>1 h incubation). All samples were analyzed in triplicate unless otherwise reported (Appendix A).

#### 3.3.4. Data Analysis

The measurement of intrinsic fluorescence intensity for each data point (i.e., taylorgram) and calculation of the apparent protein hydrodynamic radius were conducted using the Fida software (V 2.29, Fida Biosystems ApS, Copenhagen, Denmark) as previously described [11]. Viscosity compensation was applied [16]. Taylorgrams that exhibited buffer mismatch due to capillary mixing were analyzed with a two-species fit (i.e., double-Gaussian), where one species was fixed to 0.3 nm (see Appendix A for further details).

The resulting unfolding curves were fitted to the unfolding model (derived in the Appendix A):(1)$$R_{\mathrm{app}}={\left(\;\;{\begin{array}{c} {\left(R_{U}\right)}^{-1}\cdot \left(\frac{e^{\left(\frac{\Delta G{}^{\circ}\left(H_{2}O\right)+m\cdot \left[D\right]}{-R\cdot T}\right)}}{\left(e^{\left(\frac{\Delta G{}^{\circ}\left(H_{2}O\right)+m\cdot \left[D\right]}{-R\cdot T}\right)}+1\right)}\right)+{\left(R_{F}\right)}^{-1}\cdot \left(\frac{1}{\left(e^{\left(\frac{\Delta G{}^{\circ}\left(H_{2}O\right)+m\cdot \left[D\right]}{-R\cdot T}\right)}+1\right)}\right) \end{array}}^{}\right)}^{-1}$$
where *R*_app_, *R*_U_, and *R*_F_ are the apparent (measured), unfolded, and folded hydrodynamic radii, respectively. $\Delta G{}^{\circ}\left(H_{2}O\right)$ is the conformational stability in absence of denaturant, *m* is the denaturant dependency (i.e., slope) on the standard free energy change, *R* is the gas constant (1.987 × 10^−3^ kcal K^−1^ mol^−1^), *T* is the absolute temperature, and [*D*] is the denaturant concentration. 

Subsequently, the denaturation midpoint (*C*_m_) was determined from:(2)$$C_{m}=\frac{\Delta G{}^{\circ}\left(H_{2}O\right)}{m}$$

## 4. Conclusions

This work presented an optimized protocol for assessing protein stability towards unfolding, utilizing TDA employing UV-LED fluorescence detection. The TDA method only requires 40 ng of protein per data point, takes 6 min per measurement (corresponding to ~80 min for a full titration curve), has a simple workflow, is fully automated, and is conducted under native conditions in-solution. Furthermore, in contrast to existing methodologies, multiple readouts are obtained in a single-assay format, including structural features (*R*_h_), intrinsic fluorescence, and stability (i.e., *C*_m_ and Δ*G*°). We envision that the developed method can find applications within the early drug-development phase, or when only scarce amounts of material are available for analysis. 

## Figures and Tables

**Figure 1 molecules-27-02506-f001:**
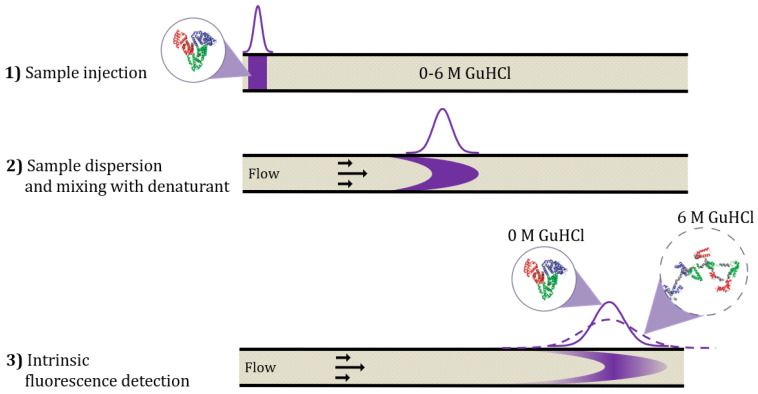
Schematics illustrating capillary mixing principle and TDA. (1) The sample is injected into a capillary filled with the denaturant solution containing from 0 to 6 M guanidinium chloride (GuHCl). (2) A laminar flow is applied, which causes dispersion of the sample into the denaturant solution. The sample and denaturant are mixed during the time course of the experiment (2–4 min). (3) The degree of sample dispersion is detected by UV-LED intrinsic fluorescence detection, which is used for calculating the hydrodynamic radius of the sample. Unfolding will be detected as an increase in hydrodynamic radius, here exemplified with 0 M GuHCl versus 6 M GuHCl. The 3D structures of native and unfolded sample (HSA) were adapted with permission from Leggio et al. [5] Copyright 2009 American Chemical Society.

**Figure 2 molecules-27-02506-f002:**
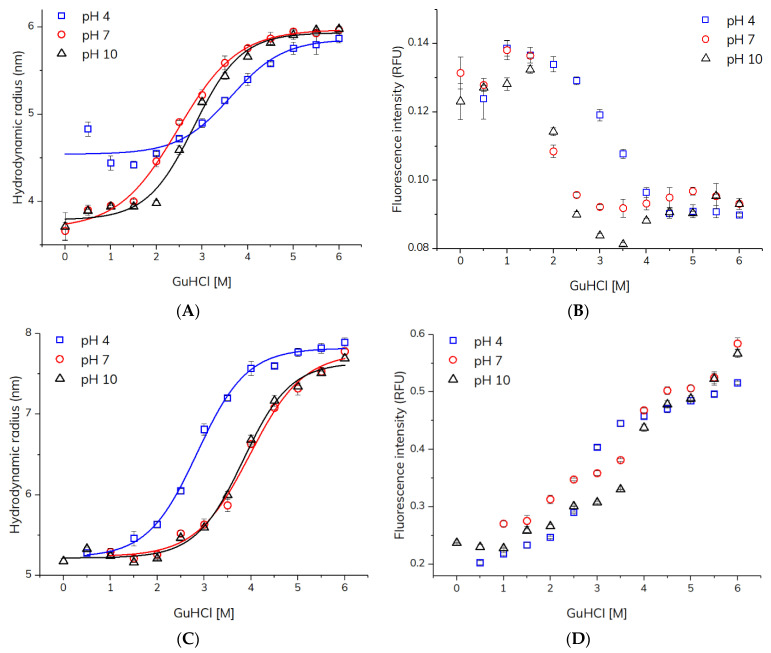
Characterization of HSA and adalimumab denaturation as a function of GuHCl concentration at pH 4.0, 7.0, and 10.0 (25 °C) using TDA with capillary mixing. (**A**) Hydrodynamic radius of HSA (1 mg/mL). (**B**) Intrinsic fluorescence intensity of HSA (1 mg/mL). (**C**) Hydrodynamic radius of adalimumab (1 mg/mL). (**D**) Intrinsic fluorescence of adalimumab (1 mg/mL). Solid lines represent fitting to the unfolding model (Equation (1)).

**Table 1 molecules-27-02506-t001:** Overview of denaturation midpoint (*C*_m_) and standard free-energy change for unfolding in absence of denaturant (Δ*G*° (H_2_O)) of adalimumab and HSA at pH 4.0, 7.0, and 10.0.

pH	Adalimumab	HAS
	*C*_m_/M [GuHCl]	
4.0	2.6	3.5
7.0	3.8	2.2
10.0	3.6	2.6
**pH**	**Δ*G*° (H_2_O) (kcal/mol)**	
4.0	3.0	3.6
7.0	3.9	2.0
10.0	4.3	3.0

## Data Availability

Raw data and data analysis are available upon request to the corresponding author Henrik Jensen at henrik@fidabio.com.

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
