# Peer review of "Quantification of Structural Integrity and Stability Using Nanograms of Protein by Flow-Induced Dispersion Analysis"

_molecules, 2022, doi:10.3390/molecules27082506_

Round 1
Reviewer 1 Report
This is a solid piece of work that demonstrates the applicability of an instrument in the field of drug development. The methodological improvement is useful to workers in the field.
I don't like the title. I would recommend removal of "fida 1" (too commercial) and replacing it with something like "new methodological approach to Flow induced dispersion analysis"
Author Response
Comments and Suggestions for Authors
This is a solid piece of work that demonstrates the applicability of an instrument in the field of drug development. The methodological improvement is useful to workers in the field.
I don't like the title. I would recommend removal of "fida 1" (too commercial) and replacing it with something like "new methodological approach to Flow induced dispersion analysis"
Reply:
We thank reviewer 1 for a positive evaluation of the manuscript. We have changed the title to:
Quantification of Structural Integrity and Stability using Nanograms of Protein by Flow Induced Dispersion Analysis
Reviewer 2 Report
Overall, the paper offers interesting practical data for using an intriguing assay method. The data analysis methodology was fairly clear, and the unfolding fitting model was explained nicely in the supporting information.
General comments:
It would be beneficial if this manuscript included slightly more background pedagogy on the TDA method. As someone lacking prior familiarity with the method, I was making assumptions about how folded and unfolded components would experience different dispersion rates that would be proportional to the diffusion rate, which would itself be a function of hydrodynamic radius. For example, in Figure 1. The legend indicates that the denatured protein has a higher hydrodynamic radius and the schematic indicates that the denatured protein would have a broader dispersion. Is this correct? I had guessed that the species with larger hydrodynamic radius would have lower dispersion?
One of the practical limitations of thermodynamic measurements such as Tm and Cm is the difficulty of applying these methods to proteins that denature irreversibly. I’d be curious about the application of this method to proteins that kind of protein.
Technical points:
When it comes to free energy values for unfolding, is there a good reason not to use SI units? kJ/mol?
“In practice, 20 μg of protein” => Given that 40 ng per measurement are needed, and 500x higher total amount is needed, perhaps the phrase in the abstract “low sample consumption (40 ng protein)” should be clarified. Perhaps, “low sample consumption (40 ng protein per data point)” or even mentioning the need for 20 μg.
Table 1. For improved clarity I recommend changing the labeling. The dash in “Cm – M [GuCl]” which is meant to separate the variable from the units can be misread as a minus.
Page 7: Line 250: The justification for having a species fixed at 0.3 nm was not clear. What is the physical meaning of a species or pseudo-species moving with Rh=0.3 nm? Also, a priori I would expect the narrow second peak to correspond to a species with high hydrodynamic radius? How does buffer mismatch create a pattern that could be interpreted as two species?
Equation 1. If there is room, it would be preferable for clarity to have the equation on one line rather than wrapper.
SI Figure 1. It would be nice to use a plot labeling scheme that works when printed in black and white.
SI page 5. “led to a shift towards higher GuHCl concentrations and thus less effective unfolding of HSA” => Is there a typo here? I would think higher GuHCl would induce more effective unfolding.
SI page 6. “the viscosity was also slight increased” => Only slightly? It would help to report the actual viscosity here. I seem to recall that 6 M GuHCl was quite viscous.
SI Figure S5: If I mobilization pressure is inversely correlated with incubation time “incubation times of 1- 26 min [50-800 mbar” should be “incubation times of 1-26 min [800-50 mbar, respectively]”
SI Figure S6: I think the word “change” in the vertical axis is probable incorrect – instead I would guess the correct axis label would be “Relative fluorescence [RFU]”
SI Equation S10: It would also be nice to avoid the line break in this equation, and perhaps to clearly label which are the four fitting parameters.
Author Response
Comments and Suggestions for Authors
Overall, the paper offers interesting practical data for using an intriguing assay method. The data analysis methodology was fairly clear, and the unfolding fitting model was explained nicely in the supporting information.
We appreciate the positive feedback from referee 2.
General comments:
It would be beneficial if this manuscript included slightly more background pedagogy on the TDA method. As someone lacking prior familiarity with the method, I was making assumptions about how folded and unfolded components would experience different dispersion rates that would be proportional to the diffusion rate, which would itself be a function of hydrodynamic radius. For example, in Figure 1. The legend indicates that the denatured protein has a higher hydrodynamic radius and the schematic indicates that the denatured protein would have a broader dispersion. Is this correct? I had guessed that the species with larger hydrodynamic radius would have lower dispersion?
An explanatory sentence (page 4, first paragraph in results and discussion) has been added describing it is mainly radial diffusion governing the dispersion.
One of the practical limitations of thermodynamic measurements such as Tm and Cm is the difficulty of applying these methods to proteins that denature irreversibly. I’d be curious about the application of this method to proteins that kind of protein.
From an experimental point of view it is also possible to measure protein unfolding irreversible. It should however be noted that the model to obtain the free energy of unfolding assumes a reversible process (two state model), and therefore not applicable to an irreversible system. We have specified (page 10 top) that the unfolding model assumes a reversible unfolding/folding process.
Technical points:
When it comes to free energy values for unfolding, is there a good reason not to use SI units? kJ/mol?
We have used the kcal/mol unit as we find it is the unit used most frequently in published data on protein unfolding. However, if journal policy is to use kJ/mol we are happy to change it.
“In practice, 20 μg of protein” => Given that 40 ng per measurement are needed, and 500x higher total amount is needed, perhaps the phrase in the abstract “low sample consumption (40 ng protein)” should be clarified. Perhaps, “low sample consumption (40 ng protein per data point)” or even mentioning the need for 20 μg.
This is a valid point – especially if only a few points are measured. In the present work the same vial is used for many measurements and whatever remaining sample can be reused in other assays. We have followed the suggestion from reviewer 2 and specified we use 40 ng pr data point.
Table 1. For improved clarity I recommend changing the labeling. The dash in “Cm – M [GuCl]” which is meant to separate the variable from the units can be misread as a minus.
We have followed the suggestion and changed from – to / in table 1.
Page 7: Line 250: The justification for having a species fixed at 0.3 nm was not clear. What is the physical meaning of a species or pseudo-species moving with Rh=0.3 nm? Also, a priori I would expect the narrow second peak to correspond to a species with high hydrodynamic radius? How does buffer mismatch create a pattern that could be interpreted as two species?
The origin of the small species is likely to be a small species present in the buffer, but not in the sample. The data analysis software allows up to 3 distinct species to be analyzed – with two species we are thus well within the capability of the instrument. Please see also discussion in the supporting information.
Equation 1. If there is room, it would be preferable for clarity to have the equation on one line rather than wrapper.
The equation is now on a single line.
SI Figure 1. It would be nice to use a plot labeling scheme that works when printed in black and white.
The figure has now been modified.
SI page 5. “led to a shift towards higher GuHCl concentrations and thus less effective unfolding of HSA” => Is there a typo here? I would think higher GuHCl would induce more effective unfolding.
The higher volume was of the protein sample – not GuHCl. This has now been specified in the text.
SI page 6. “the viscosity was also slight increased” => Only slightly? It would help to report the actual viscosity here. I seem to recall that 6 M GuHCl was quite viscous.
Indeed referee 2 is correct that 6 M GuHCl is quite viscous. For a Newtonian liquid the viscosity should be independent of the pressure (shear force). The small variation cold be due to a small non-Newtonian component which has been previously been observed for protein and polymer solutions. While this is a highly interesting point, and clearly illustrates potential for extending the methodology in this direction, it was not the main purpose of the present work to persue this aspect. We hope to be able to revisit it in a future more extensive study.
SI Figure S5: If I mobilization pressure is inversely correlated with incubation time “incubation times of 1- 26 min [50-800 mbar” should be “incubation times of 1-26 min [800-50 mbar, respectively]”
We have modified the sentence according to the suggestion from referee 2.
SI Figure S6: I think the word “change” in the vertical axis is probable incorrect – instead I would guess the correct axis label would be “Relative fluorescence [RFU]”
We have modified the figure according to the suggestion from the referee.
SI Equation S10: It would also be nice to avoid the line break in this equation, and perhaps to clearly label which are the four fitting parameters.
Equation S10 is now on a single line. The fitting variables are given by the model and also discussed in the main manuscript.
Reviewer 3 Report
This paper by Pedersen et al. is about the “Fida 1 quantifies structural integrity and stability using nanograms of protein.”
I suggest rejecting this paper as the scientific conclusions are not adequately addressed. Unfortunately, the presented paper does not meet the stringent requirements of originality, novelty and integrity.
I advise the authors to rewrite and add the answers to the questions in the paper:
Line 51: The end of the introduction, to explain: What exactly did you want to find out? What was the goal in the presented paper? Why are stability parameters not determined or at least cited for HSA and adalimumab in the presence of GuHCl?
Part 2.1., to make it more straightforward and with more details. In Part 2.1, line 55 is not clear which protein. Part 2.1 is mentioned only adalimumab; why not also HSA?
Once it is written 40 nL, other times it is written 40 ng, agree to it, or state somewhere whether it is the same. 40 ng of proteins is used for 1 pH curve, t. j. for 12 points of the curve? Please explain better.
The capillary is part of the device, is it inserted? Is the principle GuHCl titration or protein titration? How is this method significantly different from ITC? The whole principle of the method is insufficiently explained. Where is the UV-LED located? Is it part of the TDA device? I am completely lost in the presented paper.
Why were only 3 pH values (4, 7, 10) studied in the article? I recommend making the whole pH dependence, from pH 2.0 to pH 12.0. I propose to determine the stability parameters Cm and ΔG Ö¯ for both proteins for the complete pH dependence, from pH 2.0 to pH 12.0. If this is not possible, then explain why. Determine the parameters also in the presence of GuHCl. Compare the obtained data with the published data, better discuss.
Figure 2. (A) shows that point 0.5 for pH 4 is flown off the curve. I propose to add points 0.25 M, 0.75 M of GuHCl. In Figure 2 (B), adding points 1.75 and 2.25 M of GuHCl for all curves would be appropriate.
The information at lines 154-156 „Collectively, the results obtained for HSA and adalimumab unfolding in buffer solu- 154 tion as a function of GuHCl concentration and pH show that HSA is more prone to GuHCl 155 induced unfolding than adalimumab“ is new? Is it not known from previous studies? Is it not already published? If yes, please add citations.
Line 161-163: In the paper, your results agree with previous studies in the literature? How specific? In what are they the same, in what are they different? What is the new original in the presented work?
Why were these 2 proteins-adalimumab and HSA, chosen for the study? Do you plan to study other proteins?
I recommend writing a better conclusion, in which, apart from lower protein concentrations, is your study original? What are your findings? I suggest supplementing the experiments and better discussing what was found.
What are the limitations of FIDA and TDA methods?
Comments:
Figure 1 is not original, is very similar to Fig. 1 in the Paper: Pedersen et al., Scientific Reports 11, 1-10, 2021. The use of low protein concentration and given methods (FIDA and TDA) has already been in the paper: Pedersen et al., Scientific Reports 11, 1-10, 2021.
What is the novelty of this work?
Author Response
Comments and Suggestions for Authors
This paper by Pedersen et al. is about the “Fida 1 quantifies structural integrity and stability using nanograms of protein.”
I suggest rejecting this paper as the scientific conclusions are not adequately addressed. Unfortunately, the presented paper does not meet the stringent requirements of originality, novelty and integrity.
I advise the authors to rewrite and add the answers to the questions in the paper:
Line 51: The end of the introduction, to explain: What exactly did you want to find out? What was the goal in the presented paper? Why are stability parameters not determined or at least cited for HSA and adalimumab in the presence of GuHCl?
We have now specified what was the goal of the present work. The goal was to develop a new methodology for isothermal protein stability determination. For this HSA and adalimumab were used as model proteins – we are aware that they have been investigated before using other methodologies (see references such as no. 16).
Part 2.1., to make it more straightforward and with more details. In Part 2.1, line 55 is not clear which protein. Part 2.1 is mentioned only adalimumab; why not also HSA?
There is no section/part 2.1 in the manuscript we are therefore not entirely sure what the referee means here. We believe the theory and experiments are adequately explained in the manuscript as well as in the supporting information.
Once it is written 40 nL, other times it is written 40 ng, agree to it, or state somewhere whether it is the same. 40 ng of proteins is used for 1 pH curve, t. j. for 12 points of the curve? Please explain better.
For the present study these statements are equivalent as we use a protein concentration of 1 mg/mL. This has now been specified in the manuscript (result and discussion, first paragraph).
The capillary is part of the device, is it inserted? Is the principle GuHCl titration or protein titration? How is this method significantly different from ITC? The whole principle of the method is insufficiently explained. Where is the UV-LED located? Is it part of the TDA device? I am completely lost in the presented paper.
The capillary is a consumable accompanying the instrument. The basic features of the FIDA technology have been described in detail in the literature, including in the referenced paper (reference no. 9-16 and 21). ITC is a completely different technology and to our knowledge it would not be appropriate for the present investigation.
Why were only 3 pH values (4, 7, 10) studied in the article? I recommend making the whole pH dependence, from pH 2.0 to pH 12.0. I propose to determine the stability parameters Cm and ΔG Ö¯ for both proteins for the complete pH dependence, from pH 2.0 to pH 12.0. If this is not possible, then explain why. Determine the parameters also in the presence of GuHCl. Compare the obtained data with the published data, better discuss.
According to the model (please refer to the supporting information), stability parameters (the free energy of unfolding) is by definition determined in the solution not containing the chemical denaturant. In the amended manuscript the goal of the work is now further clarified (please see comment above). The main goal of the work was to develop a new flow-based methodology for convenient stability profiling of proteins requiring small amounts of sample. The two proteins used in the study serves as models to validate the new methodology. The goal was not to perform a detailed pH stability study.
Figure 2. (A) shows that point 0.5 for pH 4 is flown off the curve. I propose to add points 0.25 M, 0.75 M of GuHCl. In Figure 2 (B), adding points 1.75 and 2.25 M of GuHCl for all curves would be appropriate.
The information at lines 154-156 „Collectively, the results obtained for HSA and adalimumab unfolding in buffer solu- 154 tion as a function of GuHCl concentration and pH show that HSA is more prone to GuHCl 155 induced unfolding than adalimumab“ is new? Is it not known from previous studies? Is it not already published? If yes, please add citations.
Indeed HSA and adalimumab are certainly well-characterized proteins. The motivation of the present work has been to provide the protein research community with a new methodology that can be used to assess isothermal protein stability using very small amounts of protein. Furthermore, the novelty of the present work encompasses a single read-out that reports both hydrodynamic radius and intrinsic fluorescence during unfolding. The measured values are in good agreement with literature data as discussed in the main manuscript.
Line 161-163: In the paper, your results agree with previous studies in the literature? How specific? In what are they the same, in what are they different? What is the new original in the presented work?
Yes, there is a good agreement with results presented in reference 16 as also discussed in the manuscript.
Why were these 2 proteins-adalimumab and HSA, chosen for the study? Do you plan to study other proteins?
Adalimumab and HSA were chosen as model systems as they are examples of well-studied proteins. Further, adalimumab is an example of a therapeutic antibody and thus serves as an example of how the new methodology can be utilized in relation to protein-based therapeutics.
I recommend writing a better conclusion, in which, apart from lower protein concentrations, is your study original? What are your findings? I suggest supplementing the experiments and better discussing what was found.
We have modified the conclusion so that it now states the developed methodology is new and that combined readouts is not obtained in alternative methodologies.
What are the limitations of FIDA and TDA methods?
Hopefully the amended “introduction” and “conclusion” will now include sufficient details regarding advantages and limitations of the Fida technology.
Comments:
Figure 1 is not original, is very similar to Fig. 1 in the Paper: Pedersen et al., Scientific Reports 11, 1-10, 2021. The use of low protein concentration and given methods (FIDA and TDA) has already been in the paper: Pedersen et al., Scientific Reports 11, 1-10, 2021.
Figure 1 is not identical to the figure published in Scientific Reports. We would prefer to keep it as is.
What is the novelty of this work?
Please see comments above as well as comments from referee 1 and 2.
Round 2
Reviewer 3 Report
The author answered my questions and comments precisely, he corrected more serious errors in the manuscript. Therefore, I recommend accepting the work for publication.